# RETRACTED: TRIM10 Is Downregulated in Acute Myeloid Leukemia and Plays a Tumor Suppressive Role via Regulating NF-κB Pathway

**DOI:** 10.3390/cancers15020417

**Published:** 2023-01-08

**Authors:** Lin Li, Qi Li, Zhengrong Zou, Zoufang Huang, Yijian Chen

**Affiliations:** 1Suzhou Medical College of Soochow University, Suzhou 215123, China; 2Department of Hematology, The First Affiliated Hospital of Gannan Medical University, Ganzhou 341000, China; 3Basic Medicine Department, Chuxiong Medical and Pharmaceutical College, Chuxiong 675005, China; 4Department of Emergency, The First Affiliated Hospital of Gannan Medical University, Ganzhou 341000, China

**Keywords:** acute myeloid leukemia, TRIM10, NF-κB, tumor suppressive, epigenetic regulation

## Abstract

**Simple Summary:**

Acute myeloid leukemia (AML) remains an incurable hematological malignancy and patients have short survival due to AML relapse. In this study, we identified that TRIM10 was most downregulated in AML cell lines and AML patients. We further found that TRIM10 inhibits the growth of AML cells in vitro and in vivo. More importantly, our results indicated that TRIM10 plays a tumor suppressor role in AML cells by affecting the NF-κB signal pathway, which can be targeted with epigenetic therapy.

**Abstract:**

Background: Accumulating evidence suggests that members of the tripartite motif (TRIMs) family play a crucial role in the development and progression of hematological malignancy. Here, we explored the expression and potential role of TRIM10 in acute myeloid leukemia (AML). Methods: The expression levels of TRIM10 were investigated in AML patients and cell lines by RNA-seq, qRT-PCR and Western blotting analysis. Lentiviral infection was used to regulate the level of TRIM10 in AML cells. The effects of TRIM10 on apoptosis, drug sensitivity and proliferation of AML cells were evaluated by flow cytometry and cell-counting kit-8 (CCK-8) assay, as well as being assessed in a murine model. Results: TRIM10 mRNA and protein expression was reduced in primary AML samples and AML cell lines in comparison to the normal controls and a human normal hematopoietic cell line, respectively. Moreover, overexpression of TRIM10 in HL60 and K562 cells inhibited AML cell proliferation and induced cell apoptosis. The nude mice study further confirmed that overexpression of TRIM10 blocked tumor growth and inhibited cell proliferation. In contrast, knockdown of TRIM10 in AML cells showed contrary results. Subsequent mechanistic studies demonstrated that knockdown of TRIM10 enhanced the expression of nuclear protein P65, which implied the activation of the NF-κB signal pathway. Consistently, overexpression of TRIM10 in AML cells showed a contrary result. These data indicated that inactivation of the NF-κB pathway is involved in TRIM10-mediated regulation in AML. TRIM10 expression can be de-repressed by a combination that targets both DNA methyltransferase and histone deacetylase. Conclusions: Our results strongly suggested that TRIM10 plays a tumor suppressive role in AML development associated with the NF-κB signal pathway and may be a potential target of epigenetic therapy against leukemia.

## 1. Background

Acute myeloid leukemia (AML) is a clinically and genetically heterogeneous disease and the most common leukemia in adults [1]. AML is fatal to more than 80% of patients, especially those older than 60 years old [2,3]. Genetic abnormalities or mutations are a driving factor for the initiation, progression and relapse of AML [4,5]. For decades, chemotherapy remained unchanged, the survival improvements depend on hematopoietic stem cell transplantation (HSCT) and targeted therapy is limited for AML [2,6,7]. The persistence of residual leukemia cells is the main cause for AML relapse and they are often described as “leukemia stem cells (LSC)” [8,9]. However, leukemia relapse may also be driven by more random stress responses, which may not be specific to LSC populations [10,11]. Despite all this, AML relapse remains an enormous challenge and therapeutic strategies to eradicate leukemia cell populations are still in exploration.

Tripartite motif (TRIM) family proteins, possessing E3 ubiquitin ligase activities, are involved in cellular processes including cell growth, development, differentiation, innate immunity, autophagy and carcinogenesis [12,13]. New evidence suggests that disorder of the ubiquitin-mediated demolition of tumor suppressors or oncogene products is related to the etiology of leukemia [14]. The TRIM family, as the recognized E3 ubiquitin ligases, are vital carcinogenesis regulators [14]. Several TRIM members are involved in the acquisition of stem cell properties and the ability to maintain a stem-like phenotype [15]. Many TRIM proteins are involved in the development of hematological malignancies, acting as tumor suppressors or oncogenes [16]. TRIM19 is a tumor suppressor in acute promyelocytic leukemia (APL) [17]. Quintas-Cardama et al. found that TRIM62 expression in CD34^+^ cells of AML patients was significantly reduced and low levels were associated with shorter survival [18]. The zebrafish models demonstrate that reduction of TRIM33 disturbs hematopoiesis [19]. TRIM33 is a vital regulator of hematopoietic stem cells (HSCs) expansion and differentiation [20,21]. TRIM33 has also been found to be dysfunctional in many blood cancers and is mainly considered to have tumor suppressor activity [16], with the exception of B-cell acute lymphoblastic leukemia (B-ALL), where it is identified as an oncogene [22]. Low level of TRIM33 in multiple myeloma (MM) was found to be related to poor survival [23] and it acts as a tumor suppressor in MM [16,24]. Baranova et al. found that TRIM13 had low expression in patients with chronic lymphoblastic leukemia (CLL) at the late stage of the disease and showed the characteristics of a tumor suppressor [25]. TRIM10 is overexpressed in osteosarcoma tissues and confers cisplatin resistance on osteosarcoma cells. Guo et al. found that TRIM10 protein level is lower in systemic lupus erythematosus (SLE) patients than in healthy individuals and inhibits the expression of downstream genes in the IFN/JAK/STAT signaling pathway [26]. However, the role of TRIM10 in AML remains unknown.

The transcription factor nuclear factor kappa B (NF-κB), a protein complex, takes control of the expression of genes which participate in proliferation, apoptosis, immunity and inflammation [27]. Cytokines, such as TNFa, IL-13 and Toll-like receptor 4 (TLR-4), can activate NF-κB [28]. Furthermore, increased expression of NF-κB signaling and autocrine/paracrine cytokine signaling components are involved in the continuous NF-κB activation [29]. Basal NF-κB activity is essential for HSCs self-renewal and differentiation. Self-renewal and differentiation efficiency decreased upon loss of the NF-κB complexes p65 and p52 [30,31]. Moreover, knockout of p65 led to impaired hematopoietic stem/progenitor cells (HSPCs) engraftment and self-renewal [32,33]. It is well known that in most AML patients, NF-κB is constitutively activated, leading to resistance to apoptosis [33,34]. The chromosomal translocations or mutations leading to development of AML promote the increase of NF-κB activity [35,36]. Furthermore, high proteasome activity is often observed in AML patients, which can actively regulate NF-κB activity [37]. In addition, suppressed NF-κB has been proven to induce apoptosis in AML cells [38]. However, the potential molecular mechanisms leading to constitutive NF-κB activity remain unclear, and new therapeutic strategies based on inhibition of NF-κB activation are worth exploring.

Given the important biological function of TRIM10 in solid tumors, we tried to explore the role of TRIM10 in AML. In the present study, downregulation of TRIM10 expression was observed in primary AML cells and TRIM10 overexpression inhibited AML cell proliferation and promoted cell apoptosis. Moreover, previous studies have suggested the regulating role of TRIM10 on NF-κB activity [39]. Here we demonstrated that TRIM10 is downregulated and plays a tumor suppressor role in AML cells. More importantly, we found that the NF-κB pathway is involved in the biological effects and molecular mechanisms of TRIM10 in AML.

## 2. Materials and Methods

### 2.1. Patients and Clinical Characteristics

Bone marrow (BM) specimens and clinical data were obtained from patients who were diagnosed with AML between October 2019 and December 2020. One hundred and twenty AML patients and 30 donors without any malignant BM disorder as control [40] were enrolled in this study. Characteristics of newly diagnosed AML patients are described in Table 1. BM samples were obtained from patients with de novo AML (*n* = 120), relapsed AML (*n* = 9) and AML complete remission (CR) (*n* = 46). Ficoll-Hypaque (Sigma-Aldrich, St. Louis, MO, USA) density gradient separation was used to isolate BM mononuclear cells.

### 2.2. Mice

Female B-NDG mice (18–20 g) purchased from Jiansu Biocytogen Co., Ltd. (Nantong, China) were used [41,42,43]. Mice were housed and maintained in a specific pathogen-free environment and used when they were between 5 and 6 weeks of age. AML xenograft mouse model was established through subcutaneous injection of 5 × 10^6^ TRIM10 overexpressing HL60 cells (TRIM10-OE) or HL60 cells transfected with empty vector (Vector) into the left flank of B-NDG mice. 10 days after injection of the AML cells, when the tumors became palpable, mice were divided into the TRIM10 overexpressing group and the empty vector group randomly. Tumor size was measured every 2 days and tumor volume was calculated by the formula: 0.5 × length × width^2^ [44,45]. When the tumor volume reached 2000 mm^3^, the animals were sacrificed. Then tumors were excised and weighed.

### 2.3. Cell Lines

The human AML cell lines HL-60, THP-1, K562, K562/ADR (Cell Resource Center, Central South University, Changsha, China), the MOLM13 and MV4-11 cell lines (Cell Resource Center, Jinan University, Guangzhou, China), the human lymphocytic leukemia cell line NALM6 (Cell Bank, Chinese Academy of Sciences, Beijing, China) and the human normal hematopoietic cell line GM12878 (Cancer Research Institute, Central South University) were cultured in RPMI-1640 (Gibco, Thermo Fisher Scientific, Waltham, MA, USA) medium supplemented with 10% fetal bovine serum (Gibco, USA) at 37 °C in a 5% CO_2_ incubator.

### 2.4. Cell Proliferation Assay

HL-60 cells, HL-60 TRIM10-shRNA1 cells and HL-60 TRIM10-shRNA2 cells were seeded at a density of 2200 cells/well in 96-well plates and maintained at 37 °C in a humidified 5% CO_2_ incubator for 24 h. CCK-8 solution (Dojindo, Kumamoto, Japan) was added (10 µL per well) and was incubated for 4 h, then absorbance at 450 nm was measured on a microplate reader.

### 2.5. Cell Death Assay

The cells were harvested after treatment for 48 h, washed with ice-cold PBS, and resuspended in 400 μL binding buffer. Cells were stained with phycoerthyrin (PE)-conjugated Annexin V and 7-aminoactinomycin D (7- AAD) (BD Bioscience, San Jose, CA, USA) according to the manufacturer’s instructions. Cell death was determined on a FACScan (Becton Dickinson, Franklin Lakes, NJ, USA) by the percentages of Annexin V-positive cells.

### 2.6. Cell Cycle Analysis

Cell cycle distribution of HL-60 cells and HL-60 TRIM10-OE cells was determined using flow cytometric analysis. Cells were resuspended into 5 × 10^5^ cells/mL and stained according to the manufacturer’s instructions of the Cell Cycle Kit (US Everbright Inc, San Ramon, CA, USA). Cells were analyzed by flow cytometry immediately after staining.

### 2.7. Quantitative Real-Time PCR Assays (qRT-PCR)

TRIzol reagent (Invitrogen, Waltham, MA, USA) was used to extract total RNA. HiScript III RT SuperMix for qPCR (Vazyme, Nanjing, China) was used to synthesize complementary DNA (cDNA) with 1 μg of total RNA according to the manufacturer’s instructions. With specific primers, cDNAs were used to perform qRT-PCR analysis. qRT-PCR was performed using the LightCycler 480 real-time PCR instrument (Roche, Basel, Switzerland). All reactions were run in a two-step qRT-PCR (95 °C for 30 s, followed by 40 cycles of 95 °C for 10 s and 60 °C for 30 s) according to the manufacturer’s protocol. mRNA expression was calculated by the comparative 2^−ΔΔCT^ method, with clinical samples using ABL1 and cell lines using β-actin as the endogenous control [40]. The primers sequences are listed in the Online Appendix A.

### 2.8. Western Blot Analysis

Cells were lysed with RIPA lysis buffer (NCM Biotech, Suzhou, China) freshly supplemented with protease and phosphatase inhibitor mixture (Thermo Scientific, Waltham, MA, USA). Protein in sample buffer was electrophoresed in denaturing 10% SDS-PAGE and then transferred to polyvinylidene fluoride (PVDF) membranes (Millipore, Burlington, MA, USA). The membranes were blocked in 5% skim milk for an hour at room temperature and then incubated with primary antibodies overnight at 4 ℃. After incubation with a secondary antibody, the blots were then washed and detected with the ChemiDox XRS Chemiluminescence imaging system (Bio-Rad Laboratories, Hercules, CA, USA). Primary antibodies were as follows: anti-TRIM10 (ab151306, Abcam, Cambridge, UK, 1:1500); anti-NF-κB p65 (ab16502, Abcam, 1:1500); anti-β-actin (AF7018, Affinity Biosciences, Cincinnati, OH, USA, 1:2000). And all of the original uncropped western blot was shown in the Appendix A.

### 2.9. DNA Isolation, Bisulfite Modification and Methylation-Specific PCR (MSP)

Genomic DNA Purification Kit (Vazyme, China) was used to extract genomic DNA from BM mononuclear cells. Then the CpGenome DNA Modification Kit (Vazyme, China) was applied to modify genomic DNA followed by storage at −80 °C. Takara Taq™ Hot Start Version (Tokyo, Japan) was applied to detect TRIM10 methylation using MSP with primers presented in Appendix A. MSP reaction was performed on a LightCycler 480 real-time PCR instrument (Roche, Switzerland). The normalized ratio (N_M-TRIM10_) was applied to evaluate the level of TRIM10 methylation. N_M_-TRIM10 was calculated using the following formula: N_M-TRIM10_ = (E_M-TRIM10_)^ΔCT M-TRIM10 (control-sample)^ ÷ (E_ALU_) ^ΔCT ALU(control-sample)^.

### 2.10. Plasmid Construction and Retroviral Infection

After synthesis of the oligoduplexes, the TRIM10-specific shRNAs were cloned into the GV248.puro vector. DNA sequencing was used to verify the successful plasmid construction. The pCMV6-TRIM10 plasmid was purchased from Origene. According to the instructions, recombinant lentivirus was produced by transient transfection of HEK293T cells. For transfections, cells were seeded in a 6-well plate 24 h before the experiment. HL-60 cells were infected with lentivirus expressing TRIM10 (TRIM10-OE), TRIM10-shRNA1 and TRIM10-shRNA2 or empty vector. HL-60 TRIM10 over expression cells and TRIM10-shRNA cells were cultured with puromycin for 72 h to produce stable TRIM10 over expression and knockdown cell lines. The targeted TRIM10 sequence is as follows: TRIM10-shRNA1, GCTCCCTATAGGGAACAAATC; and TRIM10-shRNA2, GCATCCTCTTAGCACA ATTGG.

### 2.11. Luciferase Reporter Assays

The TRIM10-silenced HL60 or K562 cells and their control cells were transfected with NF-κB-Luc luciferase, TCF/LEF1-Luc and FHRE-Luc reporter plasmids using Lipofectamine 2000, respectively. After transfecting for 48 h, the cells were harvested, lysed and luciferase activity was determined using a Luciferase Assay System according to the manufacturer’s instructions. The effect of TRIM10 on the transcriptional activity of NF-κB in AML cells was then analyzed.

### 2.12. Statistical Analysis

The data were shown as mean ± standard deviation (SD). Differences among three groups were determined by analysis of Mann–Whitney U, whereas differences between two groups were evaluated by the Student’s t test. *p* values < 0.05 indicated statistical significance. Statistical analysis was performed by the SPSS 19.0 and bar graph or line charts were drawn in the GraphPad Prism 7 software. Mice were randomly assigned to groups using the random number table.

## 3. Results

### 3.1. Loss of TRIM10 Expression in AML Patients

We first explored the mRNA levels of TRIM10 in bone marrow (BM) samples from AML patients (*n* = 120) and samples from normal controls (*n* = 30). Clinical characteristics of the AML patients were summarized in Table 1 and Table 2. Results showed that expression of TRIM10 was markedly decreased in AML patients compared with normal controls (*p* < 0.001, Figure 1A). In addition, loss of TRIM10 expression was observed in patients with newly diagnosed AML (*n* = 120, *p* < 0.001) and relapsed AML (*n* = 9, *p* < 0.001) but not in patients with complete remission (*n* = 46, Figure 1B). Furthermore, compared to normal controls (*n* = 12), TRIM10 protein levels were significantly depressed in AML patients (*n* = 12, *p* < 0.001, Figure 1C,D). These results indicated that TRIM10 is generally downregulated in AML patients.

### 3.2. TRIM10 Is Downregulated in AML Cell Lines

We next examined the mRNA and protein levels of TRIM10 in AML cell lines using qRT-PCR and western blot. Compared with the control cell line GM12878, both the protein (Figure 2A,B) and mRNA (Figure 2C) levels of TRIM10 were significantly down-regulated in the six AML cell lines (*p* < 0.05). It is worth noting that TRIM10 mRNA was lower in AML cells than in the human lymphocytic leukemia cells NALM6 (*p* < 0.05, Figure 2C), however, the protein levels showed no difference (*p* > 0.05, Figure 2B). The reason may be the influence of post-translational regulation. Among AML cell lines, the adriamycin-resistant cell line K562/ADR showed lower protein (*p* = 0.047, Figure 2B) and mRNA (*p* = 0.035, Figure 2C) levels of TRIM10 than its parent cell line K562, indicating that TRIM10 may play an important role in AML drug resistance. Consistently, these data showed that TRIM10 expression is generally downregulated in AML cell lines.

### 3.3. Overexpression of TRIM10 Inhibited AML Cell Proliferation and Induced Cell Apoptosis

To further explore the function of TRIM10 in AML, we overexpressed TRIM10 using lentivirus in both the HL60 and K562 AML cell lines (Figure 3A,B). Cell viability assay, apoptosis assay and cell cycle analysis were performed. The results showed that TRIM10 overexpression significantly inhibited cell growth in both the HL60 and K562 AML cell lines (Figure 3C,D). We found more Annexin V-positive cells in the TRIM10 overexpressing HL60 and K562 AML cell lines compared with controls (Figure 3E). Cell cycle assays showed overexpression of TRIM10 increased the percentage of G_0_/G_1_ phase and decreased the percentage of G_2_M phase in HL60 and K562 cell lines (Figure 3F). These results have confirmed that overexpression of TRIM10 depresses proliferation of AML cells in vitro, so we subsequently asked whether overexpression of TRIM10 inhibits growth of AML cells in vivo. The TRIM10-overexpressing HL60 cells (TRIM10-OE group, *n* = 5 mice) and HL60 cells transfected with empty vector (Vector group, *n* = 5 mice) were injected into mice for 27 days. Figure 3G, H and I show that the TRIM10-overexpressing tumors developed in mice subcutaneously are smaller than the empty vector tumors.

### 3.4. TRIM10 Downregulation Activates the NF-κB Signalling Pathway in AML Cells

Two shRNA sequences (shRNA1 and shRNA2) were designed to knock down TRIM10 in AML cell lines. qRT-PCR and western blots confirmed that both shRNAs were effective in knocking down TRIM10 in both the HL60 and K562 AML cell lines (Figure 4A,B). Consistently, we found that knockdown of TRIM10 promoted cell growth in the HL60 and K562 AML cell lines (Appendix A). It has been generally recognized that NF-κB is constitutively activated in the cell-enriched CD34^+^ cell population in a large percentage of AML patients (33). To address the potential mechanism responsible for TRIM10-induced inhibition of growth in AML cells, we thus inquired whether the NF-κB pathway is involved in TRIM10-mediated tumor suppressive effects. We firstly analyzed the effect of TRIM10 on the expression of NF-κB p65. Western blot analysis showed that compared to the control group, the levels of NF-κB p65 were dramatically increased in both TRIM10-sh1 and TRIM10-sh2 groups in the HL60 and K562 AML cell lines (Figure 4C,D). On the contrary, the levels of NF-κB p65 were decreased in the TRIM10-overexpressing HL60 and K562 AML cell lines (Figure 4G). Subsequently, we analyzed the effect of TRIM10 on the transcriptional activity of NF-κB by luciferase reporter assays. As shown in Figure 4E,F, loss of TRIM10 drastically promoted the activation of NF-κB in AML cells (*p* < 0.01), whereas overexpression of TRIM10 suppressed the activation of NF-κB in AML cells (Figure 4H, *p* < 0.01).

### 3.5. TRIM10 Downregulation Is Associated with DNA Methylation in AML Cells

DNA hypermethylation of gene promoters is frequently observed in AML and often correlates with transcriptional repression and tumor progression [46]. To investigate the TRIM10 methylation levels in AML patients, we first examined the RQ-MSP result in 80 AML patients and 12 normal control donors. Our data showed that TRIM10 methylation level was significantly higher in AML patients compared to normal controls (*p* < 0.01, Figure 5A). DNA hypermethylation events occur frequently in AML and are generally catalyzed by DNA methyltransferases (DNMTs), including DNMT1, DNMT3A and DNMT3B [47]. DNA methylation in promoter regions is associated with changes in gene expression and silencing [48]. Therefore, we evaluated the expression of the three DNMT enzymes to validate the relationship between DNA hypermethylation and gene down-regulation. As shown in Figure 5B, gene expression levels of DNMT1, DNMT3A and DNMT3B are higher in AML patients compared to normal controls (*p* < 0.05). These results indicate that DNA hypermethylation may play a role in TRIM10 downregulation in AML patients.

### 3.6. De-Repression of TRIM10 with DNMT Inhibitor or in Combination with HDAC Inhibitor Leads to Remarkable Apoptosis in AML Cells

Previous studies suggest that epigenetic repression of gene expression in cancer might involve histone modification [49]. Thus we enquired whether the histone deacetylase (HDAC) inhibitor chidamide (Shenzhen Chipscreen Biosciences, Shenzhen, China) regulates expression of TRIM10. AML cells HL60 and K562 were treated with different doses of chidamide for 48 h, then expression levels of TRIM10 were determined with qRT-PCR. As shown in Figure 6A,B, there is no difference in the TRIM10 expression level in AML cells that were treated with chidamide versus those that were not, indicating that epigenetic repression of TRIM10 expression might involve an alternative mechanism. TRIM10 downregulation was conjectured to be related with DNA hypermethylation (Figure 5A,B), therefore we next investigated TRIM10 expression in AML cells treated with hypomethylating agents. Consistently, we found that the TRIM10 expression level was significantly increased after DNMT inhibitor azacitidine (MedChemExpress, Princeton, NJ, USA) treatment in a dose-dependent manner (Figure 6C,D). More importantly, the TRIM10 expression level was evidently increased in the combination group compared to the single agent treatment group. These results suggest that the effect on TRIM10 expression by DNMT inhibitor azacitidine is enhanced by HDAC inhibitor chidamide (Figure 6E,F). To further confirm the synergistic effect of azacitidine and chidamide on AML cells, HL60 and K562 cells were treated with 5 μM azacitidine and 1 μM chidamide singly or in combination, then an apoptosis assay was performed using flow cytometry. As shown in Figure 6G,H, the apoptotic effect of azacitidine on both HL60 and K562 cells was enhanced markedly by chidamide. These results are consistent with the previous TRIM10 expression studies.

## 4. Discussion

TRIM proteins are engaged in various biological processes of tumors cells and alterations of TRIM proteins may influence transcriptional regulation, cell proliferation and apoptosis [14,17]. In the current study, we found that TRIM10 was downregulated in primary AML cells and AML cell lines. We further demonstrated that TRIM10 inhibits cell growth by regulating NF-κB activity in AML cells (Figure 7).

The role of TRIM family members in the development and progression of blood cancer has been studied for years, among which oncogenic and tumor suppressive members have been confirmed [15,16]. Previous studies show that most TRIM proteins are acting as a tumor suppressor in hematological malignancies [16]. Our preliminary study on RNA-seq and differentially expressed analysis (DEGs) of bone marrow leukemia cells from AML patients found several significantly different low expression genes including TRIM10 (data not shown). In this study, we first determined the expression of TRIM10 mRNA and protein levels in 120 samples from patients with newly diagnosed AML and 6 AML cell lines. The results showed that the expression of TRIM10 was significantly downregulated in AML cells, indicating that it might function as a tumor suppressor in AML.

To explore biological functions of TRIM10 in AML cells, we employed a plasmid to overexpress TRIM10 in both HL60 and K562 cell lines with low endogenous TRIM10 levels. We found decreased cell growth, increased apoptosis rate, cell cycle arrest in G0/G1 phase and impaired proliferation capacity of AML cells in vitro and in vivo after TRIM10 overexpression. In addition, we generated TRIM10-knockdown cell lines using shRNAs (TRIM10 sh1 and TRIM10 sh2) in HL60 and K562 cell lines, and observed significant promotion in cell growth (Appendix A). These data suggested that overexpression of TRIM10 inhibits cell proliferation and induces cell apoptosis of AML cells. The molecular pathway by which TRIM10 inhibits AML cell proliferation was unclear.

To explore the underlying mechanism of TRIM10 on cell proliferation, we examined the expression of NF-κB p65 proteins. We found that TRIM10 overexpression decreased NF-κB p65 protein expression in both AML cell lines HL60 and K562. On the contrary, knockdown of TRIM10 increased NF-κB p65 protein expression in both HL60 and K562. In addition, downregulation of TRIM10 promoted the transcriptional activity of NF-κB in AML cells, whereas overexpression of TRIM10 suppressed the activation of NF-κB in AML cells. These results suggested that the NF-κB pathway might contribute to the biological effects of TRIM10 in AML.

The role of TRIM proteins in the leukaemogenesis regulation is complex and often cell-type specific [16,17]. The normal function of TRIM19 was missing in the APL, B-ALL and lymphoma [16]. Aucagne et al. reported that TRIM33 was reduced in 35% of chronic myelomonocytic leukaemia (CMML) patients [50]. Gatt et al. demonstrated that TRIM13 reduction resulted in decreased proliferation of MM cells, along with the NF-κB pathway and proteasome activity inhibition [51]. In this study, our data show that TRIM10 expression is generally decreased in AML cells and plays a tumor-suppressive role in AML. Though TRIM10 was recognized as an oncogenic gene in osteosarcoma cells [52], this is the first study revealing the role of TRIM10 in AML. Furthermore, we explored the underlying molecular mechanisms of biological functions of TRIM10 in AML, which might involve the NF-κB pathway and are consistent with previous studies [39,52].

Despite intensive treatment with chemotherapy and HSCT, high incidence of relapse and poor survival rate of AML remain an unresolved problem [4]. Previous studies demonstrated that downregulation of TRIM33 in blood cancers was caused by gene promoter hypermethylation and the hypomethylating agent could restore expression [50]. The DNMT inhibitors approved for older AML patients have been clinically tested in combination with HDAC inhibitors [53]. Therefore, we further explored the methylation of TRIM10 and its regulation mechanism. Our data showed that TRIM10 downregulation is associated with DNA hypermethylation in AML patients. Moreover, we found that combination treatment of AML cells with DNMT and HDAC inhibitors result in synergistic TRIM10 downregulation, which is consistent with previous studies [53]. Consistently, combination treatment with DNMT and HDAC inhibitors lead to synergistic apoptosis rate. To sum up, these data indicate that TRIM10 is a hypermethylation gene and might play a target role in the combination therapy of DNMT and HDAC inhibitors.

## 5. Conclusions

In summary, we evaluated the expression of TRIM10 in AML patients and cell lines and further explored its role. Our data show that TRIM10 exhibits tumor suppressing activity in AML. Future studies can investigate whether TRIM10 could be used as a biomarker of response to hypomethylating agents in AML. Our findings would help to better understand the role of TRIM10 in AML and highlight TRIM10 as a candidate gene for therapeutic target.

## Figures and Tables

**Figure 1 cancers-15-00417-f001:** Downregulation of TRIM10 in AML patients. (**A**) Analysis of TRIM10 mRNA expression in 120 AML patients and 30 normal controls determined by quantitative real-time PCR analysis (qRT-PCR). (**B**) Analysis of TRIM10 mRNA expression in patients with newly diagnosed AML (*n* = 120), the relapsed patients (*n* = 9) and patients with complete remission (*n* = 35). (**C**) Representative western blot analysis of TRIM10 expression in patients with AML (*n* = 12) and normal controls (*n* = 12). (**D**) Statistical analysis of TRIM10 protein expression in (**C**).

**Figure 2 cancers-15-00417-f002:** TRIM10 is downregulated in AML cell lines. (**A**) Western blot analysis of TRIM10 expression in AML cell lines. (**B**) Statistical analysis of TRIM10 protein expression in 6 AML cell lines for three experiments. (**C**) Analysis of TRIM10 mRNA expression in the 6 AML cell lines with qRT-PCR. (**D**) Pearson correlation analysis for mRNA and protein expression of TRIM10 (Pearson r = 0.918, *p* = 0.003, with F test). The human lymphocytic leukemia cell line NALM6 and the human normal hematopoietic cell line GM12878 were used as the control groups. * *p* < 0.05; ** *p* < 0.01 (Student’s *t*-test).

**Figure 3 cancers-15-00417-f003:** Overexpression of TRIM10 inhibited AML cell proliferation. (**A**,**B**) qRT-PCR and Western blots confirm an evident increase of TRIM10 in HL60 and K562 AML cell lines, with empty vector transduced in the controls. (**C**,**D**) Growth curves of HL60 empty vector, HL60 TRIM10-overexpressing (OE), K562 empty vector, and K562 TRIM10 OE AML cell lines by measuring optical density (OD) value at 450 nm using CCK-8 analysis. (**E**) Analysis of apoptosis with PE-Annexin V/7-AAD staining in HL60 TRIM10 OE, K562 TRIM10 OE and their empty vector AML cell lines after transduction for 48 h. (**F**) The percentages of cell cycle phases in HL60 TRIM10 OE, K562 TRIM10 OE and their empty vector AML cell lines. (**G**–**I**) TRIM10 overexpression exhibits AML cell inhibition in vivo: (**G**) comparison of tumor nodes between TRIM10-overexpressing HL60 cells (TRIM10-OE, *n* = 5) and HL60 cells transfected with empty vector (Vector, *n* = 5); (**H**) tumor volumes between TRIM10 and Vector groups in B-NDG mice; (**I**) tumor weight between TRIM10 and Vector groups.

**Figure 4 cancers-15-00417-f004:** TRIM10 downregulation activates the NF-κB signaling pathway in AML cells. (**A**,**B**) qRT-PCR and Western blots confirmed that TRIM10 expression was significantly decreased by TRIM10-shRNA1 or shRNA2 in the HL60 and K562 AML cell lines. (**C**,**D**) Western blot analysis of the expression levels of NF-κB p65 proteins in TRIM10-shRNA1, shRNA2 or the control shRNA in the HL60 and K562 AML cell lines. (**E**,**F**) Luciferase reporter analysis of transcriptional activity of NF-κB. The luciferase activity was expressed as a fold of HL60 or K562 control shRNA, with 1 being the value for controls. (**G**,**H**) Western blot and luciferase reporter analysis of protein level (left) and transcriptional activity (right) of NF-κB, respectively.

**Figure 5 cancers-15-00417-f005:** TRIM10 downregulation is correlated with DNA methylation in AML cells. (**A**) Real-time quantitative methylation-specific PCR (RQ-MSP) analysis of TRIM10 methylation in AML patients (*n* = 80) and normal controls (*n* = 12). (**B**) qRT-PCR analysis of DNMT1, DNMT3A and DNMT3B mRNA expression in AML patients (*n* = 120) and normal controls (*n* = 30).

**Figure 6 cancers-15-00417-f006:** Combination of azacitidine (AZA) and chidamide (CHI) results in upregulation of TRIM10 expression and remarkable apoptosis in AML cells. (**A**,**B**) qRT-PCR analysis of TRIM10 expression after treatment with different dose of CHI for 48 h. (**C**,**D**) qRT-PCR analysis of TRIM10 expression after treatment with different dose of azacitidine for 48 h. (**E**,**F**) qRT-PCR analysis of TRIM10 expression after treatment with 5 μM azacitidine and 1 μM chidamide singly or in combination for 48 h. (**G**,**H**) Annexin V-PE/7-AAD double staining analysis of HL60 and K562 cells treated with 5 μM azacitidine and/or 1 μM chidamide for 48 h. Percentages of AML cell apoptosis based on three independent experiments. * *p* < 0.05; ** *p* < 0.01 (Student’s *t*-test).

**Figure 7 cancers-15-00417-f007:** The model of our working hypothesis.

**Table 1 cancers-15-00417-t001:** Correlation of TRIM10 expression with clinical and laboratorial parameters in AML patients.

Patient’s Parameters	Total (*n* = 120)	Status of TRIM10 Expression	*p* Value
High (*n* = 60)	Low (*n* = 60)
Gender: male/female	71/49	35/25	36/24	0.990
Age, median (range)	50 (12–81)	51 (12–81)	49 (13–80)	0.648
WBC × 10^9^/L, median (range)	17.1 (0.5–263.5)	36.5 (0.7–263.5)	13.9 (0.5–131.5)	0.027
Hemoglobin g/L, median (range)	73 (35–155)	70 (36–155)	77 (35–138)	0.223
Platelets × 10^9^/L, median (range)	29 (2–400)	26 (2–263)	36 (8–400)	0.035
Median BM blasts %, (range)	74.0 (21.0–97.0)	78.5 (21.0–97.0)	66 (22.0–97.0)	<0.001
Cytogenetic risk (%)				0.002
Favorable	38 (30.2)	12 (25.0)	26 (35.4)	
Intermediate	53 (45.7)	35 (50.0)	18 (41.5)	
Adverse	14 (11.6)	10 (15.6)	4 (7.7)	
No data	15 (12.4)	3 (9.4)	12 (15.4)	
Karyotypes (%)				0.248
t(8;21)/RUNX1-RUNX1T1	18 (15.0)	12 (20.0)	6 (10.0)	
inv(16)/CBFβ-MYH11	4 (3.3)	4 (6.7)	0 (0.0)	
t(15;17)/PML-RARA	16 (13.3)	10 (16.7)	6 (10.0)	
11q23/MLL	6 (5.0)	3 (5.0)	3 (5.0)	
Normal karyotype	36 (30.0)	11 (18.3)	25 (41.7)	
Complex karyotype	5 (4.2)	0 (0.0)	5 (8.3)	
Other karyotype	20 (16.7)	8 (13.3)	12 (20.0)	
No data	15 (12.5)	12 (20.0)	3 (5.0)	
FLT3 (%)				0.022
FLT3-ITD	14 (11.7)	12 (20.0)	2 (3.3)	
FLT3-TKD	4 (3.3)	1 (1.7)	3 (5.0)	
wild	80 (66.7)	35 (58.3)	45 (75.0)	
No data	22 (18.3)	12 (20.0)	10 (16.7)	
CEBPA (%)				0.335
Single Mutation	3 (2.5)	1 (1.7)	2 (3.3)	
Double Mutation	14 (11.7)	4 (6.6)	10 (16.7)	
wild	81 (67.5)	43 (71.7)	38 (63.3)	
No data	22 (18.3)	12 (20.0)	10 (16.7)	
DNMT3A (%)				0.821
mutated	5 (4.2)	2 (3.3)	3 (5.0)	
wild	93 (77.5)	46 (76.7)	47 (78.3)	
No data	22 (18.3)	12 (20.0)	10 (16.7)	
IDH1/2 (%)				0.503
mutated	14 (11.7)	9 (15.0)	5 (8.3)	
wild	84 (70.0)	41 (68.3)	43 (71.7)	
No data	22 (18.3)	10 (16.7)	12 (20.0)	

**Table 2 cancers-15-00417-t002:** Patients’ information.

Patient’s Characteristics	Normal Control (*n* = 30)	AML-CR (*n* = 46)	AML-Relapse (*n* = 9)
Median age in years (range)	30 (21–48)	40 (18–78)	52 (23–70)
Sex (male/female)	12 (40%)/18 (60%)	22 (47.8%)/24 (52.2%)	4 (44.4%)/5 (55.6%)
Unfavorable fusion gene	-	4 (8.7%)	3 (33.3%)
Unfavorable karyotype	-	3 (6.5%)	4 (44.4%)

## Data Availability

The datasets used and/or analyzed during the current study are available from the corresponding author on reasonable request.

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
