# Peer review of "RETRACTED: TRIM10 Is Downregulated in Acute Myeloid Leukemia and Plays a Tumor Suppressive Role via Regulating NF-κB Pathway"

_cancers, 2023, doi:10.3390/cancers15020417_

Round 1
Reviewer 1 Report
The authors aim to explore the role of TRIM10, a member of the tripartite motif 21 family, in acute myeloid leukemia. The presented results are interesting: TRIM10 is down regulated in AML; its downregulation has a mechanistic impact on the NF-kB function; its expression can be modulated by hypomethylating agents. Nevertheless, it's not clear why the authors selected in particular TRIM10. In fact, other members of the TRIM family have been implied in the pathogenesis of hematological malignancies (including acute leukemia) as well; moreover, since the authors state that TRIM10 has an established role in solid tumors, I suggest that this should be discussed in a more extensive way. The only evidence provided to support this statement regards osteosarcoma. I suggest to better point out why TRIM10 could be a notable player in the context of AML.
Author Response
Dear Editor and Reviewers,
Thank you for your letter and for the reviewers’ comments on our manuscript entitled “TRIM10 is downregulated in acute myeloid leukemia and plays a tumor suppressive role via regulating NF-κB pathway” (ID: cancers-2019907). The comments are all valuable and very helpful for revising and improving the quality of our manuscript. We have studied the comments carefully and have made all the necessary corrections, which we hope will meet with your approval. The detailed point-by-point revisions are provided as follows.
Replies to the reviewer's comments:
Comment 1: Nevertheless, it's not clear why the authors selected in particular TRIM10. In fact, other members of the TRIM family have been implied in the pathogenesis of hematological malignancies (including acute leukemia) as well; moreover, since the authors state that TRIM10 has an established role in solid tumors, I suggest that this should be discussed in a more extensive way. The only evidence provided to support this statement regards osteosarcoma. I suggest to better point out why TRIM10 could be a notable player in the context of AML.
Reply 1: We are appreciated to your constructive comments and expert advice. This is a critical issue raised by the reviewer. We appreciate the reviewer for offering us constructive advice. We agree with this suggestion and add “Guo et al. found that TRIM10 protein level is lower in systemic lupus erythematosus (SLE) patients than that in healthy individuals and inhibits the expression of downstream genes in the IFN/JAK/STAT signaling pathway [26].” into the Background part. Thank you again for your constructive comments and valuable suggestions.
References:
- 26. Guo M, Cao W, Chen S, Tian R, Wang L, Liu Q, Zhang L, Wang Z, Zhao M, Lu Q, Zhu H. TRIM10 binds to IFN-α/β receptor 1 to negatively regulate type I IFN signal transduction. Eur J Immunol. 2021;51(7):1762-1773.

Reviewer 2 Report
In the present manuscript, the authors attempted at investigating the role of TRIM10 in AML, by studying the expression levels of TRIM10 in primary AML, the effects of either down-regulation and up-regulation on AML cell lines, and trying to study the molecular mechanisms underlying this role.
While the topic is very interesting, several aspects of the manuscripts needs to be revised:
1) The paper should be extensively edited for English language
2)In the introduction, the role of TRIM10 and the reason why the authors decided to focus their studies specifically on this molecule should be better explained and described
3) It would be useful to see the entire picture of the membranes of all the western blots included in the figures
4) In figure 3C, it would be clearer if the label would be cell growth. In figure 3E, apoptosis levels measured by flow cytometry are shown: firstly, the levels, although statistically significant, are about 5%, which might not be biologically relevant. Activation of the apoptosis pathway should be supported by studying the levels of apoptotic proteins, such as cleaved caspase 3 or 8 or cleaved parp. Morevoer, time points are not specified, after how many days after transduction was the apoptotis measured? Data needs better signal compensation for K562 plots.
5) in figure 4, the authors use shRNA to down-regulate trim10 in the two cell lines HL60 and K562. One aspect which is not clear is the basal expression level of these proteins in the two cell lines: in figure 2A it seems to be very low (confirmed in figure 3 A and B), while in figure 4A and B the basal level is high, could the auhtors elaborate on this? in figure 4 C and D, the basal level of NF-Kb before transduction with shRNA for trim 10 is low in both the cell lines, while in figure G and H is high before transduction, could the authors explain this?
6) in line 312, this conclusion should be softened
7) In figure 6C and D, it would be interesting to see the effect of the drugs on protein elevels next to the RNA levels. In figure 7G, flow cytometry plots showing apototic levels are reported. These levels, although statistically signifcant, needs to be supported by checking apoptotic protein levels, such as cleaved caspase 3 or 8 or cleaved parp. Perhaps the authors could check the apoptotic levels at later time points. Do they also observe differentiation of the cells after treatment, for instance upregulation of CD14, CD11 or CD15?
Author Response
Dear Editor and Reviewers,
Thank you for your letter and for the reviewers’ comments on our manuscript entitled “TRIM10 is downregulated in acute myeloid leukemia and plays a tumor suppressive role via regulating NF-κB pathway” (ID: cancers-2019907). The comments are all valuable and very helpful for revising and improving the quality of our manuscript, as well as provide important guidance for our future research. We have studied the comments carefully and have made all the necessary corrections, which we hope will meet with your approval. The detailed point-by-point revisions are provided as follows.
Replies to the reviewer's comments:
Comment 1: The paper should be extensively edited for English language.
Reply 1: The reviewer has made a very good point. We apologize for the poor language of our manuscript. As per your suggestion, we have revised the words and sentences carefully. We have now worked on language and have also involved native English speakers for language corrections. We really hope that the flow and language level have been substantially improved.
Comment 2: In the introduction, the role of TRIM10 and the reason why the authors decided to focus their studies specifically on this molecule should be better explained and described.
Reply 2: We appreciate the reviewer for offering us constructive advice. As suggested, we have added information about the reason why we studied TRIM10 into the Introduction part and the Discussion part. We believe that the publisher will displayed it perfectly during typesetting. Thank you again for your constructive comments and expert advice.
Comment 3: It would be useful to see the entire picture of the membranes of all the western blots included in the figures.
Reply 3: Appreciated for your expert guidance. As per your suggestion, we have added densitometry readings/intensity ratio of each band in all Western blot figures. In addition, we have shown all the bands with all molecular weight markers on the Western in the Supplemental Materials. We feel very grateful for your specialist review of our paper.
Comment 4: In figure 3C, it would be clearer if the label would be cell growth. In figure 3E, apoptosis levels measured by flow cytometry are shown: firstly, the levels, although statistically significant, are about 5%, which might not be biologically relevant. Activation of the apoptosis pathway should be supported by studying the levels of apoptotic proteins, such as cleaved caspase-3 or -8 or cleaved PARP. Moreover, time points are not specified, after how many days after transduction was the apoptotis measured? Data needs better signal compensation for K562 plots.
Reply 4: We are appreciated to your constructive comments and expert advice. However, we have described the results in Legends to the Figures and Results. The label OD value was known to be cell growth in various previous studies [1-3]. Apoptosis levels measured by flow cytometry are about 5% which has been confirmed in repeat experiment and are accordance with previous studies about gene overexpression [1-3]. We apologize for the inconvenience caused to you at the time of review. We apologize for not showing the apoptotic proteins cleaved caspase-3 which had been determined in our study, however, weak difference were observed between the vector and OE cells. Therefore we did not explore more apoptotic proteins, such as cleaved caspase-8 and cleaved PARP. As per your suggestion, we have add “... after transduced for 48 hours” to specify time points when apoptotis measured. We are sorry that we cannot supplement better signal compensation for K562 plots. Your advice plays a vitally important role in improving the quality of our paper, which makes us feel very grateful.
Comment 5: in figure 4, the authors use shRNA to down-regulate trim10 in the two cell lines HL60 and K562. One aspect which is not clear is the basal expression level of these proteins in the two cell lines: in figure 2A it seems to be very low (confirmed in figure 3 A and B), while in figure 4A and B the basal level is high, could the authors elaborate on this? in figure 4 C and D, the basal level of NF-Kb before transduction with shRNA for trim 10 is low in both the cell lines, while in figure G and H is high before transduction, could the authors explain this?
Reply 5: We are very grateful for your insightful comments. We apologize for the inconvenience caused to you at the time of review. However, the gene expression in the parental cell was the basic level, and when the gene was knocked down or overexpressed, the basic expression level was comparatively or relatively high or low. Specifically, different numbers of cells and different antibody concentration, and even different development intensity were used for the same parental cells in WB analysis. But the difference between control and knockdown or control and overexpression were comparable because of the same testing conditions. Therefore, the basic TRIM10 gene expression and the basal level of NF-κB were different and were incomparable in different experiments. We apologize for the inconvenience caused to you at the time of review.
Comment 6: in line 312, this conclusion should be softened.
Reply 6: Appreciated for your suggestion. As per your suggestion, we have changed the conclusion as “These results indicate that, DNA hypermethylation may play a role in TRIM10 down-regulation in AML patients.”
Comment 7: In figure 6C and D, it would be interesting to see the effect of the drugs on protein levels next to the RNA levels. In figure 7G, flow cytometry plots showing apoptotic levels are reported. These levels, although statistically significant, needs to be supported by checking apoptotic protein levels, such as cleaved caspase-3 or -8 or cleaved PARP. Perhaps the authors could check the apoptotic levels at later time points. Do they also observe differentiation of the cells after treatment, for instance upregulation of CD14, CD11 or CD15?
Reply 7: We appreciate the reviewer for offering us constructive advice. As suggested, we have It has been demonstrated that chidamide [4-5] can promote apoptosis in AML cells by increasing apoptotic protein PARP and caspase-3, and has a synergistic effect in combination with low-dose decitabine, cytarabine, daunorubicin or idarubicin [6-9]. To avoid repeated research, in this study we intend to link epigenetic drug treatment with TRIM10 gene expression. Therefore we did not make too much effort to dig deeply into apoptotic proteins. We apologize for the inconvenience caused to you at the time of review. Your advice plays a vitally important role in improving the quality of our paper, which makes us feel very grateful.
References:
- Wu X, Xia J, Zhang J, et al. Phosphoglycerate dehydrogenase promotes proliferation and bortezomib resistance through increasing reduced glutathione synthesis in multiple myeloma. Br J Haematol. 2020;190(1):52-66.
- Xu L, Wu Q, Zhou X, et al. TRIM13 inhibited cell proliferation and induced cell apoptosis by regulating NF-κB pathway in non-small-cell lung carcinoma cells. Gene. 2019; 715: 144015.
- Yu S, Li Y, Ren H, et al. PDK4 promotes tumorigenesis and cisplatin resistance in lung adenocarcinoma via transcriptional regulation of EPAS1. Cancer Chemother Pharmacol. 2021; 87(2): 207-215.
- Gong K, Xie J, Yi H, Li W. CS055 (Chidamide/HBI-8000), a novel histone deacetylase inhibitor, induces G1 arrest, ROS-dependent apoptosis and differentiation in human leukaemia cells. Biochem J. 2012 ; 443(3): 735-746.
- Li Y, Chen K, Zhou Y, et al. A New Strategy to Target Acute Myeloid Leukemia Stem and Progenitor Cells Using Chidamide, a Histone Deacetylase Inhibitor. Curr Cancer Drug Targets. 2015; 15(6): 493-503.
- Xu F, Guo H, Shi M, Liu S, Wei M, Sun K, Chen Y. A combination of low-dose decitabine and chidamide resulted in synergistic effects on the proliferation and apoptosis of human myeloid leukemia cell lines. Am J Transl Res. 2019; 11(12): 7644-7655.
- Mao J, Li S, Zhao H, Zhu Y, Hong M, Zhu H, Qian S, Li J. Effects of chidamide and its combination with decitabine on proliferation and apoptosis of leukemia cell lines. Am J Transl Res. 2018; 10(8): 2567-2578.
- Li X, Yan X, Guo W, et al. Chidamide in FLT3-ITD positive acute myeloid leukemia and the synergistic effect in combination with cytarabine. Biomed Pharmacother. 2017; 90: 699-704.
- Li Y, Wang Y, Zhou Y, et al. Cooperative effect of chidamide and chemotherapeutic drugs induce apoptosis by DNA damage accumulation and repair defects in acute myeloid leukemia stem and progenitor cells. Clin Epigenetics. 2017; 9: 83.

Reviewer 3 Report
TRIM10 is downregulated in acute myeloid leukemia and plays a tumor suppressive role via regulating NF-κB pathway
Lin Li, Qi Li, Zhengrong Zou, Zoufang Huang and Yijian Chen
MDPI Cancers
Summary: Li et. al. present their finding indicating that TRIM10 is an important tumor suppressive factor in AML which may indicate prognosis and its role in AML cells is executed via the NF-kB pathway.
The study is reasonably well conducted and the conclusions made are supported by their data. I recommend publishing this story, with minor revisions/editorial changes.
Minor Comments:
Introduction has inadequate information about TRIM10. The authors could perhaps elaborate on the gap in knowledge in the field, pertaining to TRIM10.
Line 86: What does “quality of HSCs” mean?
Line 259: What are the “different groups of mice”? Please reword this section so as to, briefly, indicate the experimental setup.
Line 270: Fig.3 legend is incomplete. G, H and I are not mentioned.
Line 321: While there are many histone posttranslational modifications, the authors chose to test a HDAC inhibitor right away -the rationale for this is not clear to me.
Line 342-348: Fig.6 legend is incomplete.
Line 355: Fig.7: The model is a little confusing as it is depicted. This could be refined - separate tracks for normal cellular state, AML cells/patients, and upon combinatorial treatment with HDACi and DNMTi - to make it more easily comestible to the readers.
Author Response
Dear Editor and Reviewers,
Thank you for your letter and for the reviewers’ comments on our manuscript entitled “TRIM10 is downregulated in acute myeloid leukemia and plays a tumor suppressive role via regulating NF-κB pathway” (ID: cancers-2019907). The comments are all valuable and provide important guidance for our future research. We have studied the comments carefully and have made all the necessary corrections, which we hope will meet with your approval. The detailed point-by-point revisions are provided as follows.
Replies to the reviewer's comments:
Comment 1: Introduction has inadequate information about TRIM10. The authors could perhaps elaborate on the gap in knowledge in the field, pertaining to TRIM10.
Reply 1: We are appreciated to your constructive comments and expert advice. We agree with this suggestion, and we have added information about TRIM10 into the Introduction part. We apologize for the inconvenience caused to you at the time of review. Thank you again for your expert advice.
Comment 2: Line 86: What does “quality of HSCs” mean?
Reply 2: We are very grateful for your insightful comments. As per your suggestion, we have replaced the “quality of HSCs” with “self-renewal and differentiation efficiency of HSCs”. Your advice plays a vitally important role in improving the quality of our paper, which makes us feel very grateful.
Comment 3: Line 259: What are the “different groups of mice”? Please reword this section so as to, briefly, indicate the experimental setup.
Reply 3: We are very grateful for your insightful comments. As per your suggestion, we have changed the sentence “The TRIM10 overexpressing HL60 and K562 AML cell lines were injected to different groups of mice for 27 days.” into “The TRIM10 overexpressing HL60 cells (TRIM10-OE group, n = 5 mice) and HL60 cells transfected with empty vector (Vector group, n = 5 mice) were injected to mice for 27 days.”. We apologize if we forgot to upload the Supplementary Materials file.
Comment 4: Line 270: Fig.3 legend is incomplete. G, H and I are not mentioned.
Reply 4: We are appreciated to your constructive comments and expert advice. As per your suggestion, we have added Figure 3G, H and I legends as “(G-I) TRIM10 overexpression exhibits AML cell inhibiting in vivo: G, comparison of tumor nodes between TRIM10 overexpressing HL60 cells (TRIM10-OE, n=5) and HL60 cells transfected with empty vector (Vector, n=5); H, tumor volumes between TRIM10 and Vector groups in B-NDG mice; I, tumor weight between TRIM10 and Vector groups.” We apologize for the negligence and appreciate you for bringing our attention to them. Thank you again for your expert advice.
Comment 5: Line 321: While there are many histone posttranslational modifications, the authors chose to test a HDAC inhibitor right away -the rationale for this is not clear to me.
Reply 5: We are appreciate your constructive comments and specialist suggestion. It is known that there are three regulatory forms of epigenetics: DNA methylation, histone methylation or acetylation modification and chromatin remodeling [1-3]. It is unclear which regulatory forms are involved in the down regulation of TRIM10 gene expression. In this section, we tried to explore the regulatory mechanism of TRIM10 transcriptional inhibition in AML from two aspects based on double epigenetic therapy [4]: DNA methylation and histone acetylation. Histone methylation or other regulatory mechanisms are involved in the down regulation of TRIM10 gene expression, e.g. micro RNA. In future, more job needs to fully clarify these possible mechanisms. We apologize for the negligence and appreciate you for bringing our attention to them.
Comment 6: Line 342-348: Fig.6 legend is incomplete.
Reply 6: We are appreciated to your constructive comments and expert advice. As per your suggestion, we have added Figure 6G and H legends as “Annexin V-PE/7-AAD double staining analysis of HL60 and K562 cells treated with 5 μM azacitidine and/ or 1 μM chidamide for 48 hours. Percentages of AML cell apoptosis based on three independent experiments.” and made corresponding revision in Line 338 and 341.
Comment 7: Line 355: Fig.7: The model is a little confusing as it is depicted. This could be refined - separate tracks for normal cellular state, AML cells/patients, and upon combinatorial treatment with HDACi and DNMTi - to make it more easily comestible to the readers.
Reply 7: We are appreciated to your constructive comments and expert advice. This is a critical issue raised by the reviewer. We agree with this suggestion and have made necessary revisions in the model (Figure 7). Thank you again for offering us constructive advice.
References:
- Dawson MA, Kouzarides T. Cancer epigenetics: from mechanism to therapy. Cell. 2012; 150(1): 12-27.
- Park JW, Han JW. Targeting epigenetics for cancer therapy. Arch Pharm Res. 2019; 42(2): 159-170.
- Hogg SJ, Beavis PA, Dawson MA, et al. Targeting the epigenetic regulation of antitumour immunity. Nat Rev Drug Discov. 2020; 19(11): 776-800.
- Blagitko-Dorfs N, Schlosser P, Greve G, Pfeifer D, Meier R, Baude A, Brocks D, Plass C, Lübbert M. Combination treatment of acute myeloid leukemia cells with DNMT and HDAC inhibitors: predominant synergistic gene downregulation associated with gene body demethylation. Leukemia. 2019;33(4):945-956.
